# LEARNING SPARSE RELATIONAL TRANSITION MODELS

**Victoria Xia**\*    **Zi Wang**\*    **Kelsey Allen**    **Tom Silver**    **Leslie Pack Kaelbling**

## ABSTRACT

We present a representation for describing transition models in complex uncertain domains using relational rules. For any action, a rule selects a set of relevant objects and computes a distribution over properties of just those objects in the resulting state given their properties in the previous state. An iterative greedy algorithm is used to construct a set of *deictic references* that determine which objects are relevant in any given state. Feed-forward neural networks are used to learn the transition distribution on the relevant objects' properties. This strategy is demonstrated to be both more versatile and more sample efficient than learning a monolithic transition model in a simulated domain in which a robot pushes stacks of objects on a cluttered table.

## 1 INTRODUCTION

Many complex domains are appropriately described in terms of sets of objects, properties of those objects, and relations among them. We are interested in the problem of taking actions to change the state of such complex systems, in order to achieve some objective. To do this, we require a *transition model*, which describes the system state that results from taking a particular action, given the previous system state. In many important domains, ranging from interacting with physical objects to managing the operations of an airline, actions have *localized* effects: they may change the state of the object(s) being directly operated on, as well as some objects that are *related* to those objects in important ways, but will generally not affect the vast majority of other objects.

In this paper, we present a strategy for learning state-transition models that embodies these assumptions. We structure our model in terms of *rules*, each of which only depends on and affects the properties and relations among a small number of objects in the domain, and only very few of which may apply for characterizing the effects of any given action. Our primary focus is on learning the *kernel* of a rule: that is, the set of objects that it depends on and affects. At a moderate level of abstraction, most actions taken by an intentional system are inherently directly parametrized by at least one object that is being operated on: a robot pushes a block, an airport management system reschedules a flight, an automated assistant commits to a venue for a meeting. It is clear that properties of these "direct" objects are likely to be relevant to predicting the action's effects and that some properties of these objects will be changed. But how can we characterize which other objects, out of all the objects in a household or airline network, are relevant for prediction or likely to be affected?

To do so, we make use of the notion of a *deictic reference*. In linguistics, a deictic (literally meaning "pointing") reference, is a way of naming an object in terms of its relationship to the current situation rather than in global terms. So, "the object I am pushing," "all the objects on the table nearest me," and "the object on top of the object I am pushing" are all deictic references. This style of reference was introduced as a representation strategy for AI systems by Agre & Chapman (1987), under the name *indexical-functional* representations, for the purpose of compactly describing policies for a video-game agent, and has been in occasional use since then.

We will learn a set of deictic references, for each rule, that characterize, relative to the object(s) being operated on, which other objects are relevant. Given this set of relevant objects, the problem

---

\*Equal contribution. Massachusetts Institute of Technology, 77 Massachusetts Ave., Cambridge, MA 02139. {`vxia,ziw,krallen,tslvr,lpk`}`@mit.edu`. We gratefully acknowledge support from NSF grants 1523767 and 1723381; from AFOSR grant FA9550-17-1-0165; from ONR grant N00014-18-1-2847; from Honda Research; from Draper Laboratory; and from the Center for Brains, Minds and Machines (CBMM), funded by NSF STC award CCF-1231216. KA acknowledges support from NSERC. Any opinions, findings, and conclusions or recommendations expressed in this material are those of the authors and do not necessarily reflect the views of our sponsors.

of describing the transition model on a large, variable-size domain, reduces to describing a transition model on fixed-length vectors characterizing the relevant objects and their properties and relations, which we represent and learn using standard feed-forward neural networks.

Next, we briefly survey related work, describe the problem more formally, and then provide an algorithm for learning both the structure, in terms of deictic references, and parameters, in terms of neural networks, of a sparse relational transition model. We go on to demonstrate this algorithm in a simulated robot-manipulation domain in which the robot pushes objects on a cluttered table.

## 2  RELATED WORK

Rule learning has a long history in artificial intelligence. The novelty in our approach is the combination of learning discrete structures with flexible parametrized models in the form of neural networks.

**Rule learning** We are inspired by very early work on rule learning by Drescher (1991), which sought to find predictive rules in simple noisy domains, using Boolean combinations of binary input features to predict the effects of actions. This approach has a modern re-interpretation in the form of schema networks (Kansky et al., 2017). The rules we learn are *lifted*, in the sense that they can be applied to objects, generally, and are not tied to specific bits or objects in the input representation and *probabilistic*, in the sense that they make a distributional prediction about the outcome. In these senses, this work is similar to that of Pasula et al. (2007) and methods that build on it ((Mourão et al., 2012), (Mourão, 2014), (Lang & Toussaint, 2010).) In addition, the approach of *learning* to use deictic expressions was inspired by Pasula et al. and used also by Marom & Rosman (2018) in the form of object-oriented reinforcement learning and by Benson (1997). Benson (1997), however, relies on a full description of the states in ground first-order logic and does not have a mechanism to introduce new deictic references to the action model. Our representation and learning algorithm improves on the Pasula et al. strategy by using the power of feed-forward neural networks as a local transition model, which allows us to address domains with real-valued properties and much more complex dependencies. In addition, our EM-based learning algorithm presents a much smoother space in which to optimize, making the overall learning faster and more robust. We do not, however, construct new functional terms during learning; that would be an avenue for future work for us.

**Graph network models** There has recently been a great deal of work on learning graph-structured (neural) network models Battaglia et al. (2018). There is a way in which our rule-based structure could be interpreted as a kind of graph network, although it is fairly non-standard. We can understand each object as being a node in the network, and the deictic functions as being labeled directed hyper-edges (between sets of objects). Unlike the typical graph network models, we do not condition on a fixed set of neighboring nodes and edges to compute the next value of a node; in fact, a focus of our learning method is to determine which neighbors (and neighbors of neighbors, etc.) to condition on, depending on the current state of the edge labels. This means that the relevant neighborhood structure of any node changes dynamically over time, as the state of the system changes. This style of graph network is not inherently better or worse than others: it makes a different set of assumptions (including a strong default that most objects do not change state on any given step and the dynamic nature of the neighborhoods) which are particularly appropriate for modeling an agent's interactions with a complex environment using actions that have relatively local effects.

## 3  PROBLEM FORMULATION

We assume we are working on a class of problems in which the domain is appropriately described in terms of objects. This method might not be appropriate for a single high-dimensional system in which the transition model is not sparse or factorable, or can be factored along different lines (such as a spatial decomposition) rather than along the lines of objects and properties. We also assume a set of primitive *actions* defined in terms of control programs that can be executed to make actual changes in the world state and then return. These might be robot motor skills (grasping or pushing an object) or virtual actions (placing an order or announcing a meeting). In this section, we formalize this class of problems, define a new rule structure for specifying probabilistic transition models for these problems, and articulate an objective function for estimating these models from data.

### 3.1 RELATIONAL DOMAIN

A problem *domain* is given by tuple $\mathcal{D} = (\Upsilon, \mathcal{P}, \mathcal{F}, \mathcal{A})$ where $\Upsilon$ is a countably infinite universe of possible objects, $\mathcal{P}$ is a finite set of properties $P_i : \Upsilon \mapsto \mathbb{R}, i \in [N_{\mathcal{P}}] = \{1, \cdots, N_{\mathcal{P}}\}$, and $\mathcal{F}$ is a finite set of deictic reference functions $F_i : \Upsilon^{m_i} \mapsto \wp(\Upsilon), i \in [N_{\mathcal{F}}]$ where $\wp(\Upsilon)$ denotes the powerset of $\Upsilon$. Each function $F_i \in \Upsilon$ maps from an ordered list of objects to a set of objects, and we define it as

$$F_i(o_1, \ldots, o_{m_i}) = \{o \mid f_i(o, o_1, \ldots, o_{m_i}) = \text{True}, \ o, o_j \in \Upsilon, \forall j \in [m_i]\} \ ,$$

where the relation $f_i : \Upsilon^{m_i+1} \mapsto \{\text{True}, \text{False}\}$ is defined in terms of the object properties in $\mathcal{P}$. For example, if we have a location property $P_{\text{loc}}$ and $m_i = 1$, we can define $f_i(o, o_1) = \mathbb{1}_{\|P_{\text{loc}}(o) - P_{\text{loc}}(o_1)\| < 0.5}$ so that the function $F_i$ associated with $f_i$ maps from one object to the set of objects that are within $0.5$ distance of its center; here $\mathbb{1}$ is an indicator function. Finally, $\mathcal{A}$ is a set of *action templates* $A_i : \mathbb{R}^{d_i} \times \Upsilon^{n_i} \mapsto \Psi, i \in [N_{\mathcal{A}}]$, where $\Psi$ is the space of executable control programs. Each action template is a function parameterized by continuous parameters $\alpha_i \in \mathbb{R}^{d_i}$ and a tuple of $n_i$ objects that the action operates on. In this work, we assume that $\mathcal{P}, \mathcal{F}$ and $\mathcal{A}$ are given.[1]

A problem *instance* is characterized by $\mathcal{I} = (\mathcal{D}, \mathcal{U})$, where $\mathcal{D}$ is a domain defined above and $\mathcal{U} \subset \Upsilon$ is a finite universe of objects with $|\mathcal{U}| = N_{\mathcal{U}}$. For simplicity, we assume that, for a particular instance, the universe of objects remains constant over time. In the problem instance $\mathcal{I}$, we characterize a *state* $s$ in terms of the concrete values of all properties in $\mathcal{P}$ on all objects in $\mathcal{U}$; that is, $s = [P_i(o_j)]_{i=1,j=1}^{N_{\mathcal{P}}, N_{\mathcal{U}}} \in \mathbb{R}^{N_{\mathcal{P}} \times N_{\mathcal{U}}} = \mathfrak{S}$. A problem instance induces the definition of its *action* space $\mathfrak{A}$, constructed by applying every action template $A_i \in \mathcal{A}$ to all tuples of $n_i$ elements in $\mathcal{U}$ and all assignments $\alpha_i$ to the continuous parameters; namely, $\mathfrak{A} = \{A_i(\alpha_i, [o_{ij}]_{j=1}^{n_i}) \mid o_{ij} \in \mathcal{U}, \alpha_i \in \mathbb{R}^{d_i}\}$.

### 3.2 SPARSE RELATIONAL TRANSITION MODELS

In many domains, there is substantial uncertainty, and the key to robust behavior is the ability to model this uncertainty and make plans that respect it. A *sparse relational transition model* (SPARE) for a domain $\mathcal{D}$, when applied to a problem instance $\mathcal{I}$ for that domain, defines a probability density function on the resulting state $s'$ resulting from taking action $a$ in state $s$. Our objective is to specify this function in terms of domain elements $\mathcal{P}, \mathcal{R}$, and $\mathcal{F}$ in such a way that it will apply to any problem instance, independent of the number and properties of the objects in its universe. We achieve this by defining the transition model in terms of a set of *transition rules*, $\mathcal{T} = \{T_k\}_{k=1}^{K}$ and a score function $C : \mathcal{T} \times \mathfrak{S} \mapsto \mathbb{N}$. The score function takes in as input a state $s$ and a rule $T \in \mathcal{T}$, and outputs a non-negative integer. If the output is 0, the rule does not apply; otherwise, the rule can predict the distribution of the next state to be $p(s' \mid s, a; T)$. The final prediction of SPARE is

$$p(s' \mid s, a; \mathcal{T}) = \begin{cases} \frac{1}{|\hat{\mathcal{T}}|} \sum_{T \in \hat{\mathcal{T}}} p(s' \mid s, a; T) & \text{if } |\hat{\mathcal{T}}| > 0 \\ \mathcal{N}(s, \Sigma_{\text{default}}) & \text{otherwise} \end{cases}, \tag{1}$$

where $\hat{\mathcal{T}} = \arg\max_{T \in \mathcal{T}} C(T, s)$ and the matrix $\Sigma_{\text{default}} = \boldsymbol{I}_{N_{\mathcal{U}}} \otimes \text{diag}([\sigma_i]_{i=1}^{N_{\mathcal{P}}})$ is the default predicted covariance for any state that is not predicted to change, so that our problem is well-formed in the presence of noise in the input. Here $\boldsymbol{I}_{N_{\mathcal{U}}}$ is an identity matrix of size $N_{\mathcal{U}}$, and $\text{diag}([\sigma_i]_{i=1}^{N_{\mathcal{P}}})$ represents a square diagonal matrix with $\sigma_i$ on the main diagonal, denoting the default variance for property $P_i$ if no rule applies. Note that the transition rules will be learned from past experience with a loss function specified in Section 3.3. In the rest of this section, we formalize the definition of transition rules and the score function.

***Transition rule*** $T = (A, \Gamma, \Delta, \phi_\theta, \boldsymbol{v}_{\text{default}})$ is characterized by an action template $A$, two ordered lists of *deictic references* $\Gamma$ and $\Delta$ of size $N_\Gamma$ and $N_\Delta$, a predictor $\phi_\theta$ and the default variances $\boldsymbol{v}_{\text{default}} = [v_i]_{i=1}^{N_{\mathcal{P}}}$ for each property $P_i$ under this rule. The action template is defined as operating on a tuple of $n$ object variables, which we will refer to as $O^{(0)} = (O_i)_{i=1}^{n}, O_i \in \mathcal{U}, \forall i$. A reference list uses functions to designate a list of additional objects or sets of objects, by making *deictic* references

---

[1]There is a direct extension of this formulation in which we encode relations among the objects as well. Doing so complicates notation and adds no new conceptual ideas, and in our example domain it suffices to compute spatial relations from object properties so there is no need to store relational information explicitly, so we omit it from our treatment.

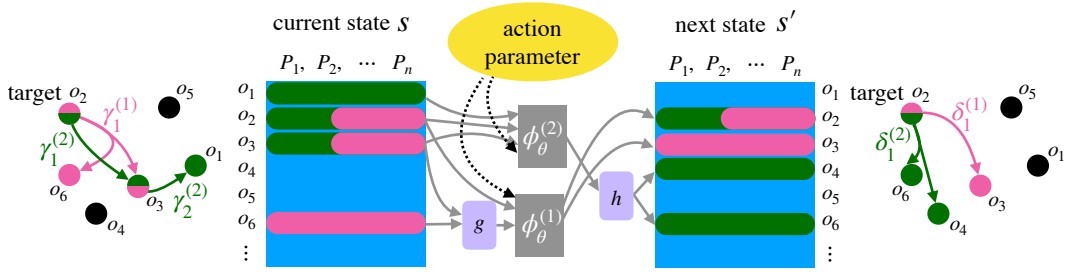

Figure 2: Instead of directly mapping from current state $s$ to next state $s'$, our prediction model uses deictic references to find subsets of objects for prediction. In the left most graph, we illustrate what relations are used to construct the input objects with two rules for the same action template, $T_1 = (A, \Gamma^{(1)}, \Delta^{(1)}, \phi_\theta^{(1)}, \boldsymbol{v}_{\text{default}}^{(1)})$ and $T_2 = (A, \Gamma^{(2)}, \Delta^{(2)}, \phi_\theta^{(2)}, \boldsymbol{v}_{\text{default}}^{(2)})$, where the reference list $\Gamma^{(1)} = [(\gamma_1^{(1)}, o_2)]$ applied a deictic reference $\gamma_1^{(1)}$ to the target object $o_2$ and added input features computed by an aggregator $g$ on $o_3, o_6$ to the inputs of the predictor of rule $T_1$. Similarly for $\Gamma^{(2)} = [(\gamma_1^{(2)}, o_2), (\gamma_2^{(2)}, o_3)]$, the first deictic reference selected $o_3$ and then $\gamma_2^{(2)}$ is applied on $o_3$ to get $o_1$. The predictors $\phi_\theta^{(1)}$ and $\phi_\theta^{(2)}$ are neural networks that map the fixed-length input to a fixed-length output, which is applied to a set of objects computed from a relational graph on all the objects, derived from the reference list $\Delta^{(1)} = [(\delta_1^{(1)}, o_2)]$ and $\Delta^{(2)} = [(\delta_1^{(2)}, o_2)]$, to compute the whole next state $s'$. Because $\delta_1^{(2)}(o_2) = (o_4, o_6)$ and the $\phi_\theta^{(2)}$ is only predicting a single property, we use a "de-aggregator" function $h$ to assign its prediction to both objects $o_4, o_6$.

based on previously designated objects. In particular, $\Gamma$ generates a list of objects whose properties affect the prediction made by the transition rule, while $\Delta$ generates a list of objects whose properties are affected after taking an action specified by the action template $A$.

We begin with the simple case in which every function returns a single object, then extend our definition to the case of sets. Concretely, for the $t$-th element $\gamma_t$ in $\Gamma$ ($t \in [N_\Gamma]$), $\gamma_t = (F, (O_{k_j})_{j=1}^m)$ where $F \in \mathcal{F}$ is a deictic reference function in the domain, $m$ is the arity of that function, and integer $k_j \in [n+t-1]$ specifies that object $O_{n+t}$ in the object list can be determined by applying function $F$ to objects $(O_{k_j})_{j=1}^m$. Thus, we get a new list of objects, $O^{(t)} = (O_i)_{i=1}^{n+t}$. So, reference $\gamma_1$ can only refer to the objects $(O_i)_{i=1}^n$ that are named in the action, and determines an object $O_{n+1}$. Then, reference $\gamma_2$ can refer to objects named in the action or those that were determined by reference $\gamma_1$, and so on.

When the function $F$ in $\gamma_t = (F, (O_{k_j})_{j=1}^m) \in \Gamma$ returns a set of objects rather than a single object, this process of adding more objects remains almost the same, except that the $O_t$ may denote sets of objects, and the functions that are applied to them must be able to operate on sets. In the case that a function $F$ returns a set, it must also specify an *aggregator*, $g$, that can return a single value for each property $P_i \in \mathcal{P}$, aggregated over the set. Examples of aggregators include the mean or maximum values or possibly simply the cardinality of the set.

For example, consider the case of pushing the bottom (block $A$) of a stack of 4 blocks, depicted in Figure 1. Suppose the deictic reference is $F = \texttt{above}$, which takes one object and returns a set of objects immediately on top of the input object. Then, by applying $F = \texttt{above}$ starting from the initial set $O_0 = \{A\}$, we get an ordered list of sets of objects $(O_0, O_1, O_2)$ where $O_1 = F(O_0) = \{B\}, O_2 = F(O_1) = \{C\}$.

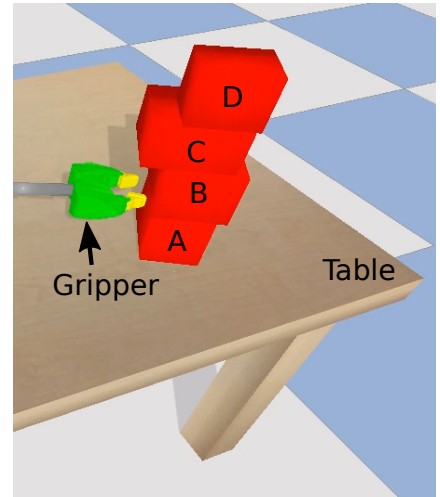

Figure 1: A robot gripper is pushing a stack of 4 blocks on a table.

Returning to the definition of a transition rule, we now can see informally that if the parameters of action template $A$ are instantiated to actual objects in a

problem instance, then $\Gamma$ and $\Delta$ can be used to determine lists of *input* and *output* objects (or sets of objects). We can use these lists, finally, to construct input and output vectors. The input vector $\boldsymbol{x}$ consists of the continuous action parameters $\alpha$ of action $A$ and property $P_i(O_t)$ for all properties $P_i \in \mathcal{P}$ and objects $O_t \in O^{N_\Gamma}$ that are selected by $\Gamma$ in arbitrary but fixed order. In the case that $O_t$ is a set of size greater than one, the aggregator associated with the function $F$ that computed the reference is used to compute $P_i(O_t)$. Similar for the desired output construction, we use the references in the list $\Delta$, initialize $\hat{O}^{(0)} = O^{(0)}$, and gradually add more objects to construct the output set of objects $\hat{O} = \hat{O}^{(N_\Delta)}$. The output vector is $\boldsymbol{y} = [P(\hat{o})]_{\hat{o} \in \hat{O}, P \in \mathcal{P}}$ where if $\hat{o}$ is a set of objects, we apply a mean aggregator on the properties of all the objects in $\hat{o}$.

The *predictor* $\phi_\theta$ is some functional form $\phi$ (such as a feed-forward neural network) with parameters (weights) $\theta$ that will take values $\boldsymbol{x}$ as input and predict a distribution for the output vector $\boldsymbol{y}$. It is difficult to represent arbitrarily complex distributions over output values. In this work, we restrict ourselves to representing a Gaussian distributions on all property values in $\boldsymbol{y}$, encoded with a mean and independent variance for each dimension.

Now, we describe how a transition rule can be used to map a state and action into a distribution over the new state. A transition rule $T = (A, \Gamma, \Delta, \phi_\theta, \boldsymbol{v}_{\text{default}})$ *applies* to a particular state-action $(s, a)$ pair if $a$ is an instance of $A$ and if none of the elements of the input or output object lists is empty. To construct the input (and output) list, we begin by assigning the actual objects $o_1, \ldots, o_n$ to the object variables $O_1, \ldots, O_n$ in action instance $a$, and then successively computing references $\gamma_i \in \Gamma$ based on the previously selected objects, applying the definition of the deictic reference $F$ in each $\gamma_i$ to the actual values of the properties as specified in the state $s$. If, at any point, a $\gamma_i \in \Gamma$ or $\delta_i \in \Delta$ returns an empty set, then the transition rule does not apply. If the rule does apply, and successfully selects input and output object lists, then the values of the input vector $\boldsymbol{x}$ can be extracted from $s$, and predictions are made on the mean and variance values $\Pr(\boldsymbol{y} \mid \boldsymbol{x}) = \phi_\theta(\boldsymbol{x}) = \mathcal{N}(\mu_{\theta_1}(\boldsymbol{x}), \Sigma_{\theta_2}(\boldsymbol{x}))$.

Let $\left(\mu_{\theta_1}^{(ij)}(\boldsymbol{x}), \Sigma_{\theta 2}^{(ij)}(\boldsymbol{x})\right)$ be the vector entry corresponding to the predicted Gaussian parameters of property $P_i$ of $j$-th output object set $\hat{o}_j$ and denote $s[o, P_i]$ as the property $P_i$ of object $o$ in state $s$, for all $o \in \mathcal{U}$. The predicted distribution of the resulting state $p(s' \mid s, a; T)$ is computed as follows:

$$p(s'[o, P_i] \mid s, a; T) = \begin{cases} \frac{1}{|\{j:o\in\hat{o}_j\}|} \sum_{\{j:o\in\hat{o}_j\}} \mathcal{N}(\mu_{\theta_1}^{(ij)}(\boldsymbol{x}), \Sigma_{\theta 2}^{(ij)}(\boldsymbol{x})) & \text{if } |\{j : o \in \hat{o}_j\}| > 0 \\ \mathcal{N}(s[o, P_i], v_i) & \text{otherwise} \end{cases}$$

where $v_i \in \boldsymbol{v}_{\text{default}}$ is the default variance of property $P_i$ in rule $T$. There are two important points to note. First, it is possible for the same object to appear in the object-list more than once, and therefore for more than one predicted distribution to appear for its properties in the output vector. In this case, we use the mixture of all the predicted distributions with uniform weights. Second, when an element of the output object list is a set, then we treat this as predicting the same single property distribution for all elements of that set. This strategy has sufficed for our current set of examples, but an alternative choice would be to make the predicted values be *changes* to the current property value, rather than new absolute values. Then, for example, moving all of the objects on top of a tray could easily specify a change to each of their poses. We illustrate how we can use transition rules to build a SPARE in Fig. 2.

For each transition rule $T_k \in \mathcal{T}$ and state $s \in \mathfrak{S}$, we assign the **score function** value to be 0 if $T_k$ does not apply to state $s$. Otherwise, we assign the total number of deictic references plus one, $N_\Gamma + N_\Delta + 1$, as the score. The more references there are in a rule that is applicable to the state, the more detailed the match is between the rules conditions and the state, and the more specific the predictions we expect it to be able to make.

### 3.3 LEARNING SPARES FROM DATA

We frame the problem of learning a transition model from data in terms of conditional likelihood. The learning problem is, given a *problem domain description* $\mathcal{D}$ and a set of experience $\mathcal{E}$ tuples, $\mathcal{E} = \{(s^{(i)}, a^{(i)}, s'^{(i)})\}_{i=1}^n$, find a SPARE $\mathcal{T}$ that minimizes the loss function:

$$\mathcal{L}(\mathcal{T}; \mathcal{D}, \mathcal{E}) = -\frac{1}{n} \sum_{i=1}^n \log \Pr(s'^{(i)} \mid s^{(i)}, a^{(i)}; \mathcal{T}) \ . \tag{2}$$

Note that we require all of the tuples in $\mathcal{E}$ to belong to the same *domain* $\mathcal{D}$, and require for any $(s^{(i)}, a^{(i)}, s'^{(i)}) \in \mathcal{E}$ that $s^{(i)}$ and $s'^{(i)}$ belong to the same problem *instance*, but individual tuples may be drawn from different problem instances (with, for example, different numbers and types of objects). In fact, to get good generalization performance, it will be important to vary these aspects across training instances.

## 4 LEARNING ALGORITHM

We describe our learning algorithm in three parts. First, we introduce our strategy for learning $\phi_\theta$, which predicts a Gaussian distribution on $\boldsymbol{y}$, given $\boldsymbol{x}$. Then, we describe our algorithm for learning reference lists $\Gamma$ and $\Delta$ for a single transition rule, which enable the extraction of $\boldsymbol{x}$ and $\boldsymbol{y}$ from $\mathcal{E}$. Finally, we present an EM method for learning multiple rules.

### 4.1 DISTRIBUTIONAL PREDICTION

For a particular transition rule $T$ with associated action template $A$, once $\Gamma$ and $\Delta$ have been specified, we can extract input and output features $\boldsymbol{x}$ and $\boldsymbol{y}$ from a given set of experience samples $\mathcal{E}$. We would like to learn the transition rule's predictor $\phi_\theta$ to minimize Eq. (2). Our predictor takes the form $\phi_\theta(\boldsymbol{x}) = \mathcal{N}(\mu_\theta(\boldsymbol{x}), \Sigma_\theta(\boldsymbol{x}))$ and a neural network is used to predict both the mean $\mu_\theta(\boldsymbol{x})$ and the diagonal variance $\Sigma_\theta(\boldsymbol{x})$. We directly optimize the negative data-likelihood loss function

$$\mathcal{L}(\theta, \Gamma, \Delta; \mathcal{D}, \mathcal{E}) = \frac{1}{n} \sum_{i=1}^{n} \left( (\boldsymbol{y}^{(i)} - \mu_\theta(\boldsymbol{x}^{(i)}))^{\mathrm{T}} \Sigma_\theta(\boldsymbol{x})^{-1} (\boldsymbol{y}^{(i)} - \mu_\theta(\boldsymbol{x}^{(i)})) + \log \det \Sigma_\theta(\boldsymbol{x}^{(i)}) \right).$$

Let $\mathcal{E}_T \in \mathcal{E}$ be the set of experience tuples to which rule $T$ applies. Then once we have $\theta$, we can optimize the default variance of the rule $T = (A, \Gamma, \Delta, \phi_\theta, \boldsymbol{v}_{\text{default}})$ by optimizing $\mathcal{L}(\{T\}; \mathcal{D}, \mathcal{E}_T)$. It can be shown that these loss-minimizing values for the default predicted variances $\boldsymbol{v}_{\text{default}}$ are the empirical averages of the squared deviations for all unpredicted objects (i.e., those for which $\phi_\theta$ does not explicitly make predictions), where averages are computed separately for each object property.

We use $\theta, \boldsymbol{v}_{\text{default}} \leftarrow \text{LEARNDIST}(\mathcal{D}, \mathcal{E}, \Gamma, \Delta)$ to refer to this learning and optimization procedure for the predictor parameters and default variance.

### 4.2 SINGLE RULE

In the simple setting where only one transition rule $T$ exists in our domain $\mathcal{D}$, we show how to construct the input and output reference lists $\Gamma$ and $\Delta$ that will determine the vectors $\boldsymbol{x}$ and $\boldsymbol{y}$. Suppose for now that $\Delta$ and $\boldsymbol{v}_{\text{default}}$ are fixed, and we wish to learn $\Gamma$. Our approach is to incrementally build up $\Gamma$ by adding $\gamma_i = (F, (O_{k_j})_{j=1}^m)$ tuples one at a time via a greedy selection procedure. Specifically, let $R_i$ be the universe of possible $\gamma_i$, split the experience samples $\mathcal{E}$ into a training set $\mathcal{E}_{train}$ and a validation set $\mathcal{E}_{val}$,

**Algorithm 1** Greedy procedure for constructing $\Gamma$.

1: **procedure** GREEDYSELECT($\mathcal{D}, \mathcal{E}, A, \Delta, N_\Gamma$)
2:     train model using $\Gamma_0 = \emptyset$, save loss $L_0$
3:     $i \leftarrow 1$
4:     **while** $i \leq N_\Gamma$ **do**
5:         $\gamma_i \leftarrow \text{None}$ ; $L_i \leftarrow \infty$
6:         **for all** $\gamma \in R_i$ **do**
7:             $\Gamma_i \leftarrow \Gamma_{i-1} \cup \{\gamma\}$
8:             $\theta, \boldsymbol{v}_{\text{default}} \leftarrow \text{LEARNDIST}(\mathcal{D}, \mathcal{E}_{train}, \Gamma_i, \Delta)$
9:             $l \leftarrow \mathcal{L}(\mathcal{T}_\gamma; \mathcal{D}, \mathcal{E}_{val})$
10:             **if** $l < L_i$ **then** $L_i \leftarrow l$ ; $\gamma_i \leftarrow \gamma$
11:         **if** $L_i < L_{i-1}$ **then** $\Gamma_i \leftarrow \Gamma_{i-1} \cup \{\gamma_i\}$ ; $i \leftarrow i+1$
12:         **else** break

and initialize the list $\Gamma$ to be $\Gamma_0 = \emptyset$. For each $i$, compute $\gamma_i = \arg\min_{\gamma \in R_i} \mathcal{L}(\mathcal{T}_\gamma; \mathcal{D}, \mathcal{E}_{val})$, where $\mathcal{L}$ in Eq. (2) evaluates a SPARE $\mathcal{T}_\gamma$ with a single transition rule $T = (A, \Gamma_{i-1} \cup \{\gamma\}, \Delta, \phi_\theta, \boldsymbol{v}_{\text{default}})$, where $\theta$ and $\boldsymbol{v}_{\text{default}}$ are computed using the LEARNDIST described in Section 4.1[2]. If the value of the loss function $\mathcal{L}(\mathcal{T}_{\gamma_i}; \mathcal{D}, \mathcal{E}_{val})$ is less than the value of $\mathcal{L}(\mathcal{T}_{\gamma_{i-1}}; \mathcal{D}, \mathcal{E}_{val})$, then we let $\Gamma_i = \Gamma_{i-1} \cup \{\gamma_i\}$ and continue. Else, we terminate the greedy selection process with $\Gamma = \Gamma_{i-1}$, since further growing

---

[2]When the rule $T$ does not apply to a training sample, we use for its loss the loss that results from having empty reference lists in the rule. Alternatively, we can compute the default variance $\Sigma_{\text{default}}$ to be the empirical variances on all training samples that cannot use rule $T$.

the list of deictic references hurts the loss. We also terminate the process when $i$ exceeds some predetermined maximum allowed number of input deictic references, $N_\Gamma$. Pseudocode for this algorithm is provided in Algorithm 1.

In our experiments we set $\Delta = \Gamma$ and construct the lists of deictic references using a single pass of the greedy algorithm described above. This simplification is reasonable, as the set of objects that are relevant to predicting the transition outcome often overlap substantially with the objects that are affected by the action. Alternatively, we could learn $\Delta$ via an analogous greedy procedure nested around or, as a more efficient approach, interleaved with, the one for learning $\Gamma$.

### 4.3 MULTIPLE RULES

Our training data in robotic manipulation tasks are likely to be best described by many rules instead of a single one, since different combinations of relations among objects could be present in different states. For example, we may have one rule for pushing a single object and another rule for pushing a stack of objects. We now address the case where we wish to learn $K$ rules from a single experience set $\mathcal{E}$, for $K > 1$. We do so via initial clustering to separate experience samples into $K$ clusters, one for each rule to be learned, followed by an EM-like approach to further separate samples and simultaneously learn rule parameters.

To facilitate the learning of our model, we will additionally learn *membership probabilities* $Z = ((z_{i,j})_{i=1}^{|\mathcal{E}|})_{j=1}^K$, where $z_{i,j}$ represents the probability that the $i$-th experience sample is assigned to transition rule $T_j$, and $\sum_{j=1}^K z_{i,j} = 1$ for all $i \in [|\mathcal{E}|]$. We initialize membership probabilities via clustering, then refine them through EM.

Because the experience samples $\mathcal{E}$ may come from different problem instances and involve different numbers of objects, we cannot directly run a clustering algorithm such as $k$-means on the $(s, a, s')$ samples themselves. Instead we first learn a single transition rule $T = (A, \Gamma, \Delta, \phi_\theta, \boldsymbol{v}_{\text{default}})$ from $\mathcal{E}$ using the algorithm in Section 4.2, use the resulting $\Gamma$ and $\Delta$ to transform $\mathcal{E}$ into $\boldsymbol{x}$ and $\boldsymbol{y}$, and then run $k$-means clustering on the concatenation of $\boldsymbol{x}$, $\boldsymbol{y}$, and values of the loss function when $T$ is used to predict each of the samples. For each experience sample, the squared distance from the sample to each of the $K$ cluster centers is computed, and membership probabilities for the sample to each of the $K$ transition rules to be learned are initialized to be proportional to the (multiplicative) inverses of these squared distances.

Before introducing the EM-like algorithm that simultaneously improves the assignment of experience samples to transition rules and learns details of the rules themselves, we make a minor modification to transition rules to obtain *mixture rules*. Whereas a probabilistic transition rule has been defined as $T = (A, \Gamma, \Delta, \phi_\theta, \boldsymbol{v}_{\text{default}})$, a mixture rule is $T = (A, \pi_\Gamma, \pi_\Delta, \Phi)$, where $\pi_\Gamma$ represents a *distribution* over all possible lists of input references $\Gamma$ (and similarly for $\pi_\Delta$ and $\Delta$), of which there are a finite number, since the set of available reference functions $\mathcal{F}$ is finite, and there is an upper bound $N_\Gamma$ on the maximum number of references $\Gamma$ may contain. For simplicity of terminology, we refer to each possible list of references $\Gamma$ as a *shell*, so $\pi_\Gamma$ is a distribution over possible shells. Finally, $\Phi = (\Gamma^{(k)}, \Delta^{(k)}, \phi_{\theta^{(k)}}, \boldsymbol{v}_{\text{default}}^{(k)})_{k=1}^\kappa$ is a collection of $\kappa$ transition rules (i.e., predictors $\phi_{\theta^{(k)}}$, each with an associated $\Gamma^{(k)}$, $\Delta^{(k)}$, and $\boldsymbol{v}_{\text{default}}^{(k)}$). To make predictions for a sample $(s, a)$ using a mixture rule, predictions from each of the mixture rule's $\kappa$ transition rules are combined according to the probabilities that $\pi_\Gamma$ and $\pi_\Delta$ assign to each transition rule's $\Gamma^{(k)}$ and $\Delta^{(k)}$. Rather than having our EM approach learn $K$ transition rules, we instead learn $K$ mixture rules, as the distributions $\pi_\Gamma$ and $\pi_\Delta$ allow for smoother sorting of experience samples into clusters corresponding to the different rules, in contrast to the discrete $\Gamma$ and $\Delta$ of regular transition rules.

As before, we focus on the case where for each mixture rule, $\Gamma^{(k)} = \Delta^{(k)}$, $k \in [\kappa]$, and $\pi_\Gamma = \pi_\Delta$ as well. Our EM-like algorithm is then as follows:

1. For each $j \in [K]$, initialize distributions $\pi_\Gamma = \pi_\Delta$ for mixture rule $T_j$ as follows. First, use the algorithm in Section 4.2 to learn a transition rule on the *weighted* experience samples $\mathcal{E}_{Z_j}$ with weights equal to the membership probabilities $Z_j = (z_{i,j})_{i=1}^{|\mathcal{E}|}$. In the process of greedily assembling reference lists $\Gamma = \Delta$, data likelihood loss function values are computed for multiple *explored* shells, in addition to the shell $\Gamma = \Delta$ that was ultimately selected. Initialize $\pi_\Gamma = \pi_\Delta$ to distribute weight proportionally, ac-

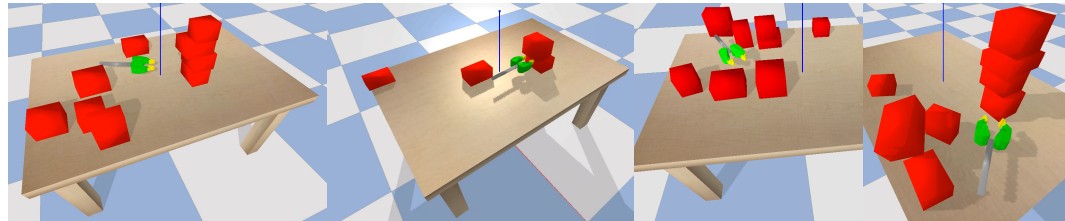

Figure 3: Representative problem instances sampled from the domain.

cording to data likelihood, for these explored shells: $\pi_\Gamma(\Gamma) = \exp(-\mathcal{L}(\mathcal{T}_\Gamma; \mathcal{D}, \mathcal{E}_{Z_j}))/\chi$, where $\mathcal{T}_\Gamma$ is the SPARE model with a single transition rule $T = (A, \Gamma, \Delta = \Gamma, \phi_\theta)$, and $\chi = (1-\epsilon)\sum_\Gamma \exp(-\mathcal{L}(\mathcal{T}_\Gamma; \mathcal{D}, \mathcal{E}_{Z_j}))$, with the summation taken over all explored shells $\Gamma$, is a normalization factor so that the total weight assigned by $\pi_\Gamma$ to explored shells is $1-\epsilon$. The remaining $\epsilon$ probability weight is distributed uniformly across unexplored shells.

2. For each $j \in [K]$, let $T_j = (A, \pi_\Gamma, \pi_\Delta, \Phi)$, where we have dropped subscripting according to $j$ for notational simplicity:

  (a) For $k \in [\kappa]$, train predictor $\Phi_k = (\Gamma^{(k)}, \Delta^{(k)}, \phi_{\theta^{(k)}}, \boldsymbol{v}_{\text{default}}^{(k)})$ using the procedure in Section 4.2 on the weighted experience samples $\mathcal{E}_{Z_j}$, where we choose $\Gamma^{(k)} = \Delta^{(k)}$ to be the list of references with $k$-th highest weight according to $\pi_\Gamma = \pi_\Delta$.

  (b) Update $\pi_\Gamma = \pi_\Delta$ by redistributing weight among the top $\kappa$ shells according to a voting procedure where each training sample "votes" for the shell whose predictor minimizes the validation loss for that sample. In other words, the $i$-th experience sample $\mathcal{E}^{(i)}$ votes for mixture rule $v(i) = k$ for $k = \arg\min_{k \in [\kappa]} \mathcal{L}(\Phi_k; \mathcal{D}, \mathcal{E}^{(i)})$. Then, shell weights are assigned to be proportional to the sum of the sample weights (i.e., membership probability of belonging to this rule) of samples that voted for each particular shell: the number of votes received by the $k$-th shell is $V(k) = \sum_{i=1}^{|\mathcal{E}|} \mathbb{1}_{v(i)=k} \cdot z_{i,j}$, for indicator function $\mathbb{1}$ and $k \in [\kappa]$. Then, $\pi_\Gamma(k)$, the current $k$-th highest value of $\pi_\Gamma$, is updated to become $V(k)/\xi$, where $\xi$ is a normalization factor to ensure that $\pi_\Gamma$ remains a valid probability distribution. (Specifically, $\xi = (\sum_{k=1}^\kappa \pi_\Gamma(k))/(\sum_{k=1}^\kappa V(k))$.)

  (c) Repeat Step 2a, in case the $\kappa$ shells with highest $\pi_\Gamma$ values have changed, in preparation for using the mixture rule to make predictions in the next step.

3. Update membership probabilities by scaling by data likelihoods from using each of the $K$ rules to make predictions: $z_{i,j} = z_{i,j} \cdot \exp(-\mathcal{L}(T_j; \mathcal{D}, \mathcal{E}^{(i)}))/\zeta$, where $\exp(-\mathcal{L}(T_j; \mathcal{D}, \mathcal{E}^{(i)}))$ is the data likelihood from using mixture rule $T_j$ to make predictions for the $i$-th experience sample $\mathcal{E}^{(i)}$, and $\zeta = \sum_{j=1}^K z_{i,j} \cdot \exp(-\mathcal{L}(T_j; \mathcal{D}, \mathcal{E}^{(i)}))$ is a normalization factor to maintain $\sum_{j=1}^K z_{i,j} = 1$.

4. Repeat Steps 2 and 3 some fixed number of times, or until convergence.

## 5 EXPERIMENTS

We apply our approach, SPARE, to a challenging problem of predicting pushing stacks of blocks on a cluttered table top. We describe our domain, the baseline that we compare to and report our results.

### 5.1 OBJECT MANIPULATION DOMAIN

In our domain $\mathcal{D} = (\Upsilon, \mathcal{P}, \mathcal{F}, \mathcal{A})$, the object universe $\Upsilon$ is composed of blocks of different sizes and weight, the property set $\mathcal{P}$ includes shapes of the blocks (width, length, height) and the position of the block ($(x, y, z)$ location relative to the table). We have one action template, $push(\alpha, o)$, which pushes toward a *target object* $o$ with parameters $\alpha = (x_g, y_g, z_g, d)$, where $(x_g, y_g, z_g)$ is the 3D position of the gripper before the push starts and $d$ is the distance of the push. The orientation of the gripper and the direction of the push are computed from the gripper location and the target object

location. We simulate this 3D domain using the physically realistic PyBullet (Coumans & Bai, 2016–2018) simulator. In real-world scenarios, an action cannot be executed with the exact action parameters due to the inaccuracy in the motor and hence in our simulation, we add Gaussian noise on the action parameters during execution to imitate this effect.

We consider the following deictic references in the reference collection $\mathcal{F}$: (1) *identity* (O), which takes in an object $O$ and returns $O$; (2) *above* (O), which takes in an object $O$ and returns the object immediately above $O$; (3) *below* (O), which takes in an object $O$ and returns the object immediately below $O$; (4) *nearest* (O), which takes in an object $O$ and returns the object that is closest to $O$.

## 5.2 BASELINE METHODS

**Neural network (NN)**  We compare to a neural network function approximator that takes in as input the current state $s \in \mathbb{R}^{N_\mathcal{P} \times N_\mathcal{U}}$ and action parameter $\alpha \in \mathbb{R}^{N_\mathcal{A}}$, and outputs the next state $s' \in \mathbb{R}^{N_\mathcal{P} \times N_\mathcal{U}}$. The list of objects that appear in each state is ordered: the target objects appear first and the remaining objects are sorted by their poses (first sort by $x$ coordinate, then $y$, then $z$).

**Graph NN**  We compare to a fully connected graph NN. Each node of the graph corresponds to an object in the scene, and the action $\alpha$ is concatenated to the state of each object. Bidirectional edges connect every node in the graph. The graph NN consists of encoders for the nodes and edges, propagation networks for message passing, and a node decoder to convert back to predict the mean and variance of the next state of each object.

## 5.3 RESULTS

**Effects of deictic rules**  As a sanity check, we start from a simple problem where a gripper is pushing a stack of three blocks with two extra blocks on the table. We randomly sampled 1250 problem instances by drawing random block shapes and locations from a uniform distribution within a range while satisfying the condition that the stack of blocks is stable and the extra blocks do not affect the push. In each problem instance, we uniformly randomly sample the action parameters and obtain the training data, a collection of tuples of state, action and next state, where the target object of the push action is always the one at the bottom of the stack. We held out 20% of the training data as the validation set. We found that our approach is able to reliably select the correct combinations of the references that select all the blocks in the problem instance to construct inputs and outputs. In Fig. 4(a), we show how the performance varies as deictic references are added during a typical run of this experiment. The solid purple curves show training performance, as measured by data likelihood on the validation set, while the dashed purple curve shows performance on a held-out test set with 250 unseen problem instances. As expected, performance improves noticeably from the addition of the first two deictic references selected by the greedy selection procedure, but not from the 4th. The brown curve shows the learned default standard deviations, used to compute data likelihoods for features of objects not explicitly predicted by the rule. As expected, the learned default standard deviation drops as deictic references are added, until it levels off after the third reference is added since at that point the set of references captures all moving objects in the scene.

**Sensitivity analysis on the number of objects**  We compare our approach to the baselines in terms of how sensitive the performance is to the number of objects that exist in the problem instance. We continue the setting where a stack of three blocks lie on a table, with extra blocks that may affect the prediction of the next state. Figure 4(b) shows the performance, as measured by the log data likelihood, as a function of the number of extra blocks. For each number of extra blocks, we used 1250 training problem instances with 20% as the validation set and 250 testing problem instances. When there are no extra blocks, SPARE learns a single rule whose $\boldsymbol{x}$ and $\boldsymbol{y}$ contain the same information as the inputs and outputs for the baselines. As more objects are added to the table, NN's performance drops as the presence of these additional objects appear to complicate the scene and NN is forced to consider more objects when making its predictions. SPARE outperforms graph NN, as the good predictions for the extra blocks contribute to the log data likelihood.

Note that, performance aside, NN is limited to problems for which the number of objects in the scenes is fixed, as the it requires a fixed-size input vector containing information about all objects. Our SPARE approach does not have this limitation, and could have been trained on a single, large

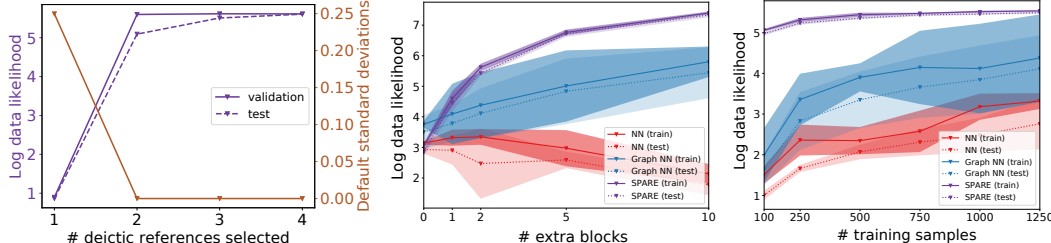

Figure 4: (a) In a simple 3-block pushing problem instance, data likelihood and learned default standard deviation both improve as more deictic references are added. (b) Comparing performance as a function of number of distractors with a fixed amount of training data. (c) Comparing sample efficiency of SPARE to the baselines. Shaded regions represent $95\%$ confidence interval.

dataset that is the combination of the datasets with varying numbers of extra objects. However, we did not do this in our experiments for the sake of providing a more fair comparison against NN.

**Sample efficiency**   We evaluate our approach on more challenging problem instances where the robot gripper is pushing blocks on a cluttered table top and there are two additional blocks on the table that do not interfere or get affected by the pushing action. Fig. 4(c) plots the data likelihood as a function of the number of training samples. We evaluate with training samples varying from $100$ to $1250$ and in each setting, the test dataset has $250$ samples. Both our approach and the baselines benefit from having more training samples, but our approach is much more sample efficient and achieves good performance within only $500$ training samples.

**Learning multiple transition rules**   Now we put our approach in a more general setting where multiple transition rules need to be learned for prediction of the next state. Our approach adopts an EM-like procedure to assign each training sample its distribution on the transition rules and learn each transition rule with re-weighted training samples. First, we construct a training dataset and $70\%$ of it is on pushing 4-block stack. Our EM approach is able to concentrate to the 4-block case as shown in Fig. 5(a).

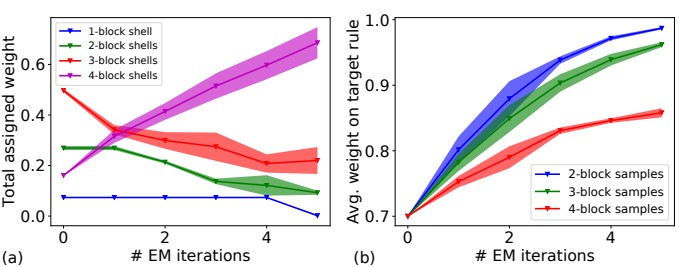

Figure 5: (a) Shell weights per iteration of our EM-like algorithm. (b) Membership probabilities of training samples per iteration.

Fig. 5(b) tracks the assignment of samples to rules over the same five runs of our EM procedure. The three curves correspond to the three stack heights in the original dataset, and each shows the average weight assigned to the "target" rule among samples of that stack height, where the target rule is the one that starts with a high concentration of samples of that particular height. At iteration 0, we see that the rules were initialized such that samples were assigned 70% probability of belonging to specific rules, based on stack height. As the algorithm progresses, the samples separate further, suggesting that the algorithm is able to separate samples into the correct groups.

**Conclusion**   These results demonstrate the power of combining relational abstraction with neural networks, to learn probabilistic state transition models for an important class of domains from very little training data. In addition, the structural nature of the learned models will allow them to be used in factored search-based planning methods that can take advantage of sparsity of effects to plan efficiently.

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

## A    DISCUSSIONS AND FUTURE WORK

In this work, we made an attempt to use deictic references to construct transition rules that can generalize over different problem instances with varying numbers of objects. Our approach can be viewed as constructing "convolutional" layers for a neural network that operates on objects. Similar to the convolutional layer operating on different sizes of images, we use deictic rules to find relations among objects and construct features for learning without a constraint on the scale of our problem or how many objects there are. Our approach is to find structures in existing data and learn the construction of deictic rules, each of which uses a specific strategy to filter out objects that are important to an action.

Many possible improvements and extensions can be made to this work, including experimentation with different template structures, more powerful deictic references, more refined feature selection, and integration with planning.

This work focused on using deictic references to select input and output *objects* for a predictor. A natural extension is to select not just objects, but also *features* of objects to obtain more refined templates and even smaller predictors, especially in cases where objects have many different properties, only some of which are relevant to the prediction problem at hand. There is also room for experimentation with different types of aggregators for combining features of objects when deictic references select multiple objects in a set.

Table 1: Sample separation from discrete clustering approach. Standard deviations are reported in parentheses.

|  |  | Num objects in sample | | |
|---|---|---|---|---|
|  |  | 2 | 3 | 4 |
|  | 1 | **0.829 (0.042)** | 0.022 (0.030) | 0.089 (0.126) |
| Cluster index | 2 | 0.115 (0.081) | **0.751 (0.028)** | 0.038 (0.025) |
|  | 3 | 0.056 (0.039) | 0.227 (0.012) | **0.872 (0.102)** |

Finally, the end purpose of obtaining a transition model for robotic actions is to enable planning to achieve high-level goals. Due to time constraints, this work did not assess the effectiveness of learned template-based models in planning, but this is a promising area of future work as the assumption that features of objects for which a template makes no explicit prediction do not change meshes well with STRIPS-style planners, as they make similar assumptions.

## B  MORE EMPIRICAL RESULTS

We here provide details on our experiments and more results in experiments.

**Experimental details**  Our experiments on SPARE in this paper used nueral network predictors for making mean predictions and variance predictions, described in Section 4.1. Each network was implemented as a fully-connected network with two hidden layers of 64 nodes each in Keras, used ReLU activations between layers, and the Adam optimizer with default parameters. Predictors for the templates approach were trained for 1000 epochs each with a decaying learning rate starting at 1e-2 and decreasing by a factor of 0.6 every 100 epochs. The baseline NN predictor was implemented in exactly the same way.

For the GNN, we used a node encoder and edge encoder to map to latent spaces of 16 dimensions. The propagation networks consisted of 2 fully connected layers of 16 units each, and the decoder mapped back to 6 dimensions: 3 for the mean, and 3 for the variance. The GNN was trained using a decaying learning rate starting at 1e-2, and decreasing by a factor of 0.5 every 100 epochs. A total of 900 epochs were used.

States were parameterized by the $(x, y, z)$ pose of each object in the scene, ordered such that the target object of the action always appeared first, and other objects appeared in random order (except for the baseline). Action parameters included the $(x, y, z)$ starting pose of the robotic gripper, as well as a "push distance" parameter that controls how long the push action lasts. Actions were implemented to be stochastic by adding some amount of noise to the target location of each push, potentially reflecting inaccuries in robotic control.

**Initialization of Membership Probabilities**  We use the clustering-based approaches for initializing membership probabilities presented in Section 4.3. In this section, we how well our clustering approach performs.

Table 1 shows the sample separation achieved by the discrete clustering approach, where samples are assigned solely to their associated clusters found by $k$-means, on the push dataset for stacks of varying height. Each column corresponds to the one-third of samples which involve stacks of a particular height. Entries in the table show the proportion of samples of that stack height that have been assigned to each of the three clusters, where in generating these data the clusters were ordered so that the predominantly 2-block sample cluster came first, followed by the predominantly 3-block cluster, then the 4-block cluster. Values in parentheses are standard deviations across three runs of the experiment. As seen in the table, separation of samples into clusters is quite good, though 22.7% of 3-block samples were assigned to the predominantly 4-block cluster, and 11.5% of 2-block samples were assigned to the predominantly 3-block cluser.

The sample separation evidenced in Table 1 is enough such that templates trained on the three clusters of samples reliably select deictic references that consider the correct number of blocks, i.e., the 2-block cluster reliably learns a template which considers the target object and the object above the target, and similiarly for the other clusters and their respective stack heights. However, initializing

Table 2: Sample separation from clustering-based initialization of membership probabilities, where probabiliites are assigned to be proportional to inverse distance to cluster centers. Standard deviations are reported in parentheses.

|  |  | Num objects in sample | | |
|---|---|---|---|---|
|  |  | 2 | 3 | 4 |
|  | 1 | **0.595 (0.010)** | 0.144 (0.004) | 0.111 (0.003) |
| Cluster index | 2 | 0.259 (0.005) | **0.551 (0.007)** | 0.286 (0.005) |
|  | 3 | 0.146 (0.006) | 0.305 (0.003) | **0.602 (0.008)** |

Table 3: Sample separation from clustering-based initialization of membership probabilities, where probabiliites are assigned to be proportional to squared inverse distance to cluster centers. Standard deviations are reported in parentheses.

|  |  | Num objects in sample | | |
|---|---|---|---|---|
|  |  | 2 | 3 | 4 |
|  | 1 | **0.730 (0.009)** | 0.065 (0.025) | 0.118 (0.125) |
| Cluster index | 2 | 0.149 (0.079) | **0.665 (0.029)** | 0.171 (0.056) |
|  | 3 | 0.121 (0.076) | 0.270 (0.005) | **0.716 (0.069)** |

samples to belong solely to a single cluster, rather than initializing membership probabilities, is unlikely to be robust in general, so we turn to the proposed clustering-based approaches for initializing membership probabilities instead.

Table 2 is analogous to Table 1 in structure, but shows sample separation results for sample membership probabilities initialized to be proportional to the inverse distance from the sample to each of the cluster centers found by $k$-means. Table 3 is the same, except with membership probabilities initialized to be proportional to the *square* of the inverse distance to cluster centers.

Sample separation is better in the case of squared distances than non-squared distances, but it's unclear whether this result generalizes to other datasets. For our specific problem instance, the log data likelihood feature turns out to be very important for the success of these clustering-based initialization approaches. For example, Table 4 shows results analogous to those in Table 3, where the only difference is that all log data likelihoods were multiplied by five before being passed as input to the $k$-means clustering algorithm. Comparing the two tables, this scaling of data likelihood to become relatively more important as a feature results in better data separation. This suggests that the relative importance between log likelihood and the other input features is a parameter of these clustering approaches that should be tuned.

**Effect of object ordering on baseline performance** The single-predictor baseline used in our experiments receives all objects in the scene as input, but this leaves open the question of in what order these objects should be presented. Because the templates approach has the target object of the action specified, in the interest of fairness this information is also provided to the baseline by having the target object always appear first in the ordering. As there is in general no clear ordering for the remainder of the objects, we could present them in a random order, but perhaps sorting the objects according to position (first by $x$-coordinate, then $y$, then $z$) could result in better predictions than if objects are completely randomly ordered.[3]

---

[3]As this suspicion turns out to be true, the baseline experiments order blocks in sorted order by position.

Table 4: Sample separation from clustering-based initialization of membership probabilities, where probabiliites are assigned to be proportional to squared inverse distance to cluster centers, and log data likelihood feature used as part of $k$-means clustering has been multiplied by a factor of five. Standard deviations are reported in parentheses.

|  |  | Num objects in sample | | |
|---|---|---|---|---|
|  |  | 2 | 3 | 4 |
|  | 1 | **0.779 (0.017)** | 0.020 (0.001) | 0.016 (0.001) |
| Cluster index | 2 | 0.174 (0.014) | **0.744 (0.006)** | 0.118 (0.014) |
|  | 3 | 0.047 (0.005) | 0.236 (0.005) | **0.866 (0.015)** |

To analyze the effect of object ordering on baseline performance, we run the same experiment where a push action is applied to the bottom-most of a stack of three blocks, and there exist some number of additional objects on the table that do not interfere with the action in any way. Figure 6 shows our results. We test three object orderings: random ("none"), sorted according to object position ("xtheny"), and an ideal ordering where the first three objects in the ordering are exactly the three objects in the stack ordered from bottom up ("stack"). As expected, in all cases, predicted log likelihood drops as more extra blocks are added to the scene, and the random ordering performs worst while the ideal ordering performs best.

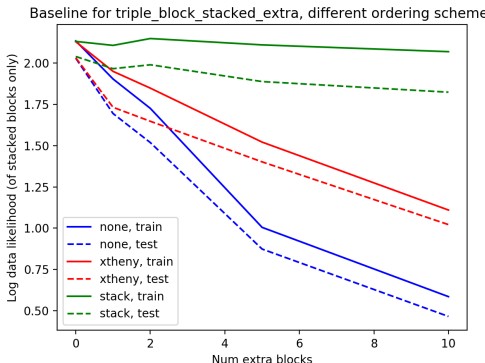

Figure 6: Effect of object ordering on baseline performance, on task of pushing a stack of three blocks on a table top, where there are extra blocks on the table that do not interfere with the push.

