# OpenReview forum: "Learning sparse relational transition models"
_ICLR.cc/2019/Conference_

### Official Review · AnonReviewer3 · 2018-11-02
**L EARNING SPARSE RELATIONAL TRANSITION MODELS**

**Rating:** 8
**Confidence:** 3

**Review:**


SUMMARY:
This work is about learning state-transition models in complex domains represented as sets of objects, their properties, ``"deictic" reference functions between sets of objects, and possible actions (or action templates). A parametric model for the actions is assumed, and these parameters act on a neural net that learns the transition model (probabilistic rule) from the current state to the next one.  It is basically this nonlinear transition model implemented by a network which makes this work different from previous models described in the literature. The relational transition model proposed is sparse, based on the assumption that actions have only ``local effects on related objects. The prediction model itself is basically a Gaussian distribution whose mean and variances are represented by neural nets. For jointly learning multiple rules, a clustering strategy is presented which assigns experience samples to transition rules. The method is applied to simulated data in the context of predicting pushing stacks of blocks on a cluttered table top.

EVALUATION:
The type of problems addressed in this paper is challenging and highly relevant for solving problems in the ``real'' world. Although the method proposed is in some sense a direct generalization of the work in [Pasula et al.], it still contains many novel and interesting aspects.Any single part of the model (like the use of Gaussians parametrized by functions implemented via neural nets) is somehow ``standard in deep latent variable models, but in complex real-world rule-learning problems the whole system presented  defines  certainly a big improvement over the state-of-the-art, which in my opinion has the potential to indeed advance this field of research.

---

### Official Review · AnonReviewer1 · 2018-11-02
**An approach to learn lifted transition rules using neural networks that take advantage of relational structure**

**Rating:** 7
**Confidence:** 2

**Review:**

An approach is proposed that learns transition rules in terms of local contexts. Specifically, transition rules make predictions as a distribution over the set of possible states based on local context of objects. A learning algorithm is described that learns the transition rules by maximizing the conditional likelihood. To learn the rules jointly with selecting the right samples for the transition rule, and EM algorithm is proposed.

The paper is well-written. The contribution seems significant considering that relational structure is integrated with neural networks in a systematic manner. Though written from the perspective of learning transition rules for tasks such as robotic manipulation, I think similar ideas can be for general tasks that can benefit from both relational structure and neural network representation.  Learning lifted rules has also been studied in  domains such as ILP and Statistical Relational Learning (Getoor and Taskar 07)(lifted rules with uncertainty). I think including their perspective and commenting on their relationship with the proposed work will be useful.

Experiments are performed on a robotic manipulation task involving pushing a stack of blocks in a cluttered environment. A method that does not take object relations into account and simply predicts the state transition is used as baseline for comparison. The proposed approach shows the benefits of exploiting the structure between objects. There is not too much discussion on scalability. Does the propose method scale up for learning transition rules in real tasks? Are there any tradeoffs involved, etc. would be good to know.
In summary, this seems to be a well-written and novel contribution.

---

### Official Review · AnonReviewer2 · 2018-11-05
**Learning sparse relational transition models**

**Rating:** 6
**Confidence:** 4

**Review:**

In the manuscript "Learning sparse relational transition models", the authors combine neural nets with relational models, using ideas from linguistics. They apply this to learning the representations of the space in which a simulated robot operates in a reinforcement learning ML paradigm. This work is of interest to the AI community and ICLR is a good venue for this work.

The authors apply their model in particular to a problem in which the simulated robot must rearrange objects in space, and they achieve reasonable accuracy.

Major points:

- Organisationally, I thought that the authors could have gotten to the loss function sooner, as much of the development of the theory is lacking in motivation until specific tasks are defined.

- The application domain seemed to lose some of the power of the linguistic analysis they were doing to develop the representation through "properties" and "action templates". These definitions were quite general, but it was unclear if more than a few (with few parameters) were used in the actual application, and so it's unclear that so much generality was required by the application.

- The authors could have compared with more modern deep learning techniques for reinforcement learning such as DeepMimic (Peng et al 2018).

Minor points:
- Typesetting periods "Pasula et al. and" -> "Pasula et al.\ and"

- Page 2: "value of a note" -> "value of a node"

- 3.1 was hard to follow.

---

### Author Response · Authors · 2018-11-17
**Author rebuttal**

We thank the reviewers for their constructive feedback and address individual questions below.

AR2:

Q: "the authors could have gotten to the loss function sooner."
A: We will add an explanation near Eq. (1)  and emphasize that the transition will be learned.

Q: ... "it was unclear if more than a few (with few parameters) were used in the actual application, and so it's unclear that so much generality was required by the application."
A: It is true that, although the framework is quite general, our demonstration domain is relatively simple.  This work is not intended to be a solution to just that domain, but to introduce a new representation and learning strategy for transition models.

Q: compare to "modern deep learning techniques for reinforcement learning such as DeepMimic (Peng et al 2018)."
A: Our method aims to learn a compact transition model that can generalize to different problem instances while deep RL approaches such as DeepMimic typically focuses on obtaining a policy in a model-free manner without explicitly predicting a transition model. Hence, we can't directly compare to them. Our approach can potentially be combined with model-based deep RL approaches, thought it is not a focus of this work.

We will fix the typos.

AR1:

Q: relation to ILP and statistical relational learning (SRL)
A: We will amplify our discussion of the work of Benson (1997), which developed an inductive concept learning approach based on ILP to learn descriptions of the action model from data. Benson (1997), however, relies on a full description of the states in ground first-order logic and does not have a mechanism to introduce new "references" to the action model. Our approach is also more flexible than Benson (1997) because of the usage of neural nets as predictors. The Pasula et al work is a type of SRL, and is the closest work in that area.

Q: scalability of the approach
A: We emphasize that the rule learning EM approach can be done offline. The online prediction only requires a forward pass on the learned predictor for each applicable rule.

---

### Meta-Review · Area_Chair1 · 2018-11-06
**Good paper and valuable direction**

**Confidence:** 3
**Recommendation:** Accept (Poster)

**Metareview:**

pros:
- the paper is well-written and precise
- the proposed method is novel
- valuable for real-world problems

cons:
- Reviewer 2 expresses some concern about the organization of the paper and over-generality in the exposition
- There could be more discussion of scalability